# Measuring the Impact of the Pandemic on Female and Male Students' Learning in a Society in Transition: A Must for Sustainable Education

**Maura A. E. Pilotti** [1,*] **, Omar J. El-Moussa** [2] **and Hanadi M. Abdelsalam** [1]

1   College of Sciences and Human Studies, Prince Mohammad Bin Fahd University,
    Al Khobar 31952, Saudi Arabia; habdelsalam@pmu.edu.sa
2   Department of Student Affairs, Prince Mohammad Bin Fahd University, Dhahran 31952, Saudi Arabia;
    omar@pmu.edu.sa
*   Correspondence: maura.pilotti@gmail.com or mpilotti@pmu.edu

**Abstract:** A sustainable education amid a disruptive event (e.g., a pandemic) requires the objective assessment of learning before and during the event and, if necessary, evidence-driven solutions in response to deficiencies. The present action research study illustrates an evidence-based response of educators to the widespread concerns that learning in college students, accustomed to face-to-face courses, might have been damaged during the pandemic by the switch to the online mode. It focuses on general education (GE) courses as they usually enroll students at the beginning of their journey in higher education, and thus, a population that is likely to be particularly sensitive to unforeseen changes. Pass/fail grades in courses taught face-to-face and online synchronously by the same instructors were examined. It was hypothesized that if the switch from face-to-face to online instruction changed the students' approaches to learning, course performance would differ between the instructional modes. Differences in female and male students' adaptation responses were expected to be reflected in their course performance. The study found that female students performed better online than face-to-face in Arabic Culture, Natural Science, Math, and Wellness courses. Male students also performed better online in Math and Natural Science courses, whereas they exhibited better performance face-to-face in Arabic Culture, Wellness, and Professional Competency courses. It was concluded that basic indices of uneven performance can guide further analyses into the sources of female and male students' approaches to instructional modes.

**Keywords:** online learning; sustainable education; gender; general education; action research

## 1. Introduction

Action research in teaching is the practice of disciplined inquiry conducted by educators to inform as well as improve outcomes [1]. In contrast to traditional research, whose main goal is to generate valid knowledge, often under controlled conditions, to improve a given field of inquiry, the goal of action research is to generate knowledge that is both valid and impactful (i.e., vital to the wellbeing of the individuals it assesses) [2]. As such, the present study illustrates educators' responses to concerns regarding college students' learning during the pandemic. In a world where opinions often count more than facts, and facts may be misconstrued to serve particular self-interests, we argue that evidence-based inquiry is the most sensible first response to circulating narratives of damage. In our study, the problem-solving goal of action research is satisfied by an evaluation of the sources of any discrepancies between face-to-face (before the pandemic) and online (during the pandemic) instruction, which can then guide a surgical intervention intended to rectify discrepancies.

A key aspect of sustainable education is its ability to preserve quality while withstanding unforeseen, potentially disruptive environmental conditions [3,4]. Although the drastic measures of social distancing (such as home confinement, reduced mobility, and

widespread online instruction), used at the inception of the pandemic, have devolved into more relaxed measures, the pandemic is still present in the everyday life of college students. On most college campuses, for instance, some classes have yet to return to the face-to-face mode, due to the insufficient classroom space arising from social distancing requirements. Thus, even at these later stages of the pandemic, it is important to assess whether educators' concerns regarding college students' learning during the pandemic are supported by concrete evidence. In the present study, learning in general education (GE) courses is examined for two key reasons. Namely, GE courses not only offer the foundations for major-related coursework but also enroll students at the beginning of their educational endeavors (i.e., freshmen and sophomores) who may be particularly susceptible to the changes in instruction [5–7] brought about by the pandemic. For these students, college adjustment usually entails coping with the changes to their everyday lives related to the demands of academic work and the socio-cultural environment of the institution they selected for their education [8,9]. The pandemic adds to the burden of the changes with which these students are required to cope in their pursuit of academic success [10].

## 2. GE Curriculum

Courses in the GE curriculum are not merely mandatory courses that college students must complete to graduate [11]. They offer the foundational knowledge and skills that a student needs to succeed in the field of the chosen major. As such, they complement specialization and career preparation courses by covering a variety of topics (e.g., psychology, natural sciences, history, and culture), skills (e.g., reading, writing, speaking, and computing), and modes of inquiry (e.g., critical thinking methods) thought to be necessary for students to become educated citizens [12] and to be successful in the professions of their choice [13]. They are also thought to offer remedial opportunities to students from underserved populations whose unequal academic outcomes have endured despite increased access to higher education [14]. As such, GE curricula are seen as a critical step for ensuring that higher education institutions meet the needs of all students. Not surprisingly, performance in GE courses often predicts academic success at graduation in the selected major [15–17].

## 3. Concerns Regarding Students' Learning

During the pandemic, the unending debate regarding the quality of online learning relative to face-to-face learning has escalated in intensity and reach [18]. The term "emergency remote teaching" has been used to underscore that the sudden change to online instruction, experienced by students mostly accustomed to face-to-face instruction, was not an attempt "to re-create a robust educational ecosystem but rather to provide temporary access to instruction and instructional supports in a manner that is quick to set up and is reliably available during an emergency or crisis" [19]. This mindset has led to the argument that "well-planned online learning experiences are meaningfully different from courses offered online in response to a crisis or disaster" [19] (p. 1).

Yet, the more commonly used term "online learning" has dominated educators' expressions of concern regarding the instruction offered to students forced to remain at home. Concerns have emphasized an unwavering commitment to quality education and the need for quality control. Evidence, albeit scant, from action research suggests that regardless of the hasty temporal adoption of the online mode, teaching quality has remained a priority during the pandemic, leading to fruitful cooperations of administrators and faculty members to create viable, and hopefully, sustainable means of delivering quality education [20–23]. Not surprisingly, the experience of online teaching has been perceived by many as an opportunity to acquire digital knowledge and skills to develop a well-balanced integration of physical and digital devices and instructional methods, with the ultimate goal of creating a sustainable ecosystem for active and meaningful learning [24].

## 4. Institutional Responses

Institutions have differed in the extent to which they have enforced a unified choice of the online mode of instruction or have allowed educators to select the mode they prefer. Irrespective of the mode selected by an institution and its faculty, differences have also emerged in the platforms used for online delivery, and the degree to which institutions have supported their faculty and students through training and IT support services.

Initial institutional responses to the pandemic's imperative need for social distance have been to move courses online, opting for a synchronous, asynchronous, or blended (i.e., bichronous) instructional mode. In the synchronous mode, the regular brick-and-mortar classroom becomes virtual, thereby allowing for real-time interactions via text, voice, and video functions, while ensuring that physical distance exists between each attendee and all others. If learners cannot attend or need reiteration, records of lectures, presentations, and class discussions, are usually available for review. In the asynchronous mode, real-time interactions for lectures and class discussions are not available. Exchanges occur via discussion boards, forums, and email. Teaching materials, usually including text, and audio and video recordings, are posted online so that learners can work through them on their own time. The blended mode is intended to minimize the challenges of the asynchronous option arising from its lack of immediacy in human interactions. It combines the two previous modes so that some class activities can be carried out at the convenience of the learners (asynchronous components), whereas other activities require learners to participate in real-time (synchronous components). The amount of synchronous and asynchronous learning may vary with the content of a course and the demands its learning objectives prescribe.

According to the social-cognitive framework, learning is situated in specific contexts, which demand different levels of self-regulation [25–27]. Effort regulation, time management, concentration, and strategies to avoid distraction tend to be positively associated with academic performance in both face-to-face [28] and online courses [29,30]. Yet, the latter usually place higher self-regulation demands on the learner [31], in addition to presenting technological challenges and demanding adaptation to a suboptimal social environment, often inducing a sense of isolation, all of which have the potential to disrupt learning [32,33].

Evidence concerning comparisons of students' academic success (as measured by grades) between face-to-face (pre-pandemic) and online (during the pandemic) have yielded mixed results [20,21,23,34,35], thereby suggesting the need for individual higher education institutions and, most importantly, individual educators, to assess the status of students' learning in the courses they have offered. Yet, the extant evidence has shown that institutional support, through training in the use of technology and instructional methods as well as modeling, is key to students' academic success [20,34]. In the present study, we focus on an institution that has mandated the adoption of the synchronous online mode to all educators and ensured preparation through modeling as well as IT and instructional training. Notwithstanding institutional support, educators expressed concerns about online learning for their students who had been accustomed to face-to-face learning since elementary school. In our study, we describe the institution's assessment of the validity of such concerns, which can serve as a simple blueprint for other institutions to determine whether indeed online instruction has damaged learning. Our study focuses on GE courses because such courses are the foundations for later specialized learning. Furthermore, they enroll freshmen and sophomores, ostensibly students who may be particularly vulnerable to the effects of unforeseen instructional changes in the academic environment to which they are still adjusting. The study asks whether there are differences in enrollment and performance (as measured by pass/fail grades) between face-to-face courses taught by the same instructors before and during the pandemic. As the study takes place in a society in transition from a patriarchal order based on tribal networks [36] to one in which gender equity is gradually being inserted into its social fabric, education system, and workforce [18,37–39], gender is examined to assess whether it differentiates students'

responses to online instruction. In this socio-cultural context, women's newly acquired rights and opportunities may make them more determined to seek academic success [40], thereby expressing no diminished enrollment and learning or even growth. Alternatively, unforeseeable changes requiring a return to home confinement may make women less likely to adapt well to online instruction [41], thereby curtailing their enrollment and learning.

The present study is guided by the following hypotheses regarding the impact of the pandemic on learning:

**Hypothesis 1.** *Differences in the enrollment between face-to-face and online courses will indicate students' responses to change: fleeing versus withstanding change.*

**Hypothesis 2.** *Differences in performance between face-to-face and online courses will indicate the extent to which learning might have been affected by the switch to the online mode demanded by the pandemic.*

**Hypothesis 3.** *Gender differences in both enrollment and performance indices will illustrate the extent to which the male and female students of a society in transition may exhibit distinct responses to sudden and unforeseen change.*

## 5. Method

### 5.1. The Sample

Seven GE undergraduate courses were selected that (a) had been taught by the same instructors face-to-face before the pandemic and synchronously online during the pandemic, and (b) had sufficient enrollment to meet inferential statistics' provisions. The pass/fail grades of the undergraduate students who completed such courses ($n$ = 56,316) were obtained from the Office of the Registrar for three semesters before the pandemic (2018 and 2019) and three semesters during the pandemic (2020 and 2021). To measure pre-existing trends for male and female students, 6 semesters before the pandemic (2015, 2016, and 2017) were also included ($n$ = 50,704). The data obtained from the Office of the Registrar were anonymized to ensure compliance with the guidelines for educational research of the Office for Human Research Protections of the U.S. Department of Health and Human Services and with the American Psychological Association's ethical standards. The study was conducted under the purview of the Deanship of Research.

The educational institution targeted for the present study is a private university offering engineering, computer science, business, architecture, and law programs, located in the Eastern Region of Saudi Arabia. Although coeducational, instruction is gender-segregated with courses offered separately to female and male students. In this context, GE courses serve as foundations for the instruction of major-specific undergraduate curricula. The courses chosen for the present study pertained to the following categories: Arabic Culture (4 courses covering Arabic and Islamic customs), Communication (4 courses dedicated to written and spoken communication in English), Mathematics (9 courses covering Calculus, Statistics, and Algebra), Professional Competencies (3 courses intended to foster the development of basic skills common to a variety of professional endeavors, such as teamwork, leadership, and critical thinking), Natural Sciences (6 courses including Chemistry and Physics), Wellness (2 courses dedicated to wellness education), and Self-Assessment (3 courses devoted to the appraisal of one's general and major-specific competencies). Other courses were excluded for low enrollment and/or for not having matching instructors for the face-to-face and online modes (e.g., Biology, Economics, Foreign Languages, History, Psychology, and Sustainability).

At the institution chosen for the present study, GE courses are usually taken by freshman or sophomore students, whose ages range from 18 to 25. The specific distribution of GE courses selected by individual students depends on majors' requirements. Each course is offered to females and males separately due to gender-segregation requirements. English is the primary means of communication. Students are classified as

Arabic–English bilingual speakers with English serving as the second language (as measured by standardized tests prior to admission). Academic programs, including the GE curriculum, are accredited by the Saudi Ministry of Education and, depending on the subject matter, by specific foreign higher education entities to ensure compliance with the demands of a global economy. For instance, the GE curriculum is accredited by the Texas International Education Consortium (TIEC).

*5.2. Modes of Instruction*

All the selected courses were taught exclusively face-to-face before the pandemic and through the synchronous online mode during the pandemic. Internal surveys by the Office of the Registrar confirmed that before the pandemic students were unaccustomed to online learning. In both modes, Blackboard was used as the platform to post course materials, submit assignments, take tests, and display performance records. During the pandemic, Blackboard Collaborate was added to it to allow for real-time classes through video, audio, whiteboard, and chat functions. Although Blackboard was used in both instructional modes for test administration and the submission of assignments, guidelines for online testing required that students activate the camera and microphone functions of Blackboard Collaborate as well as use a lock-down screen. As a further measure against misconduct, plagiarism software was available to instructors to scan the content of tests and assignments.

Institutional support offered to educators included individualized training addressing technological changes and pedagogical challenges in transitioning to online instruction. In addition to institutional support, informally developed support networks among colleagues were intended to foster educators' sense of self-efficacy in a time of crisis. Specific guidelines were issued for both online instruction and assessment, which then were monitored for compliance. The guidelines for instruction focused on fostering active learning and engagement, both key principles of the selected university's pedagogy, despite the physical distance between the instructors and classmates. Additionally, guidelines dictated that tests and assignments covered five of the six levels of the Bloom taxonomy (i.e., remember, understand, apply, analyze, and evaluate) [42]. The upper level (creation) was not contemplated in light of the introductory nature of GE courses.

## 6. Results and Discussion

*6.1. Assessment of Enrollment*

First, enrollment changes were examined (see Table 1) across time as they might indicate pre-existing trends. Timeframes considered were organized into "remote pre-pandemic period" (six semesters including 2015, 2016, and 2017 academic years), "recent pre-pandemic period" (three semesters immediately before the pandemic, including 2018 and 2019), and "pandemic period" (three semesters during the pandemic, including 2020 and 2021). As noted earlier, all the courses preceding the pandemic were face-to-face. For each category of courses, a chi-squared test was performed to determine whether enrollment rates varied as a function of the instructional timeframe between female and male students. A Bonferroni correction was applied to avoid experiment-wise alpha, thereby treating as significant all tests with $p < 0.007$.

In Table 1, the last column highlights relevant shifts in enrollment between male and female students. Specifically, in Self-Assessment ($\chi^2(2, n = 13{,}348) = 116.92, p < 0.001$), Arabic culture ($\chi^2(2, n = 21{,}531) = 89.46, p < 0.001$), Professional Competencies ($\chi^2(2, n = 16{,}411) = 38.74, p < 0.001$), and Mathematics ($\chi^2(2, n = 14{,}351) = 64.18, p < 0.001$), a decline in enrollment of female students from the remote pre-pandemic period to the recent pre-pandemic period persisted during the pandemic (FMM). Instead, Communication ($\chi^2(2, n = 19{,}977) = 51.61, p < 0.001$) and Wellness ($\chi^2(2, n = 11{,}325) = 124.53, p < 0.001$), displayed a return to the enrollment of the remote pre-pandemic period during the pandemic (FMF). Namely, female students had greater enrollment than male students during the pandemic (online courses) as they also did during the remote pre-pandemic period. Natural Sciences ($\chi^2 = 6.17$, ns) did not

exhibit a significant gender-related change in enrollment as a function of time period. In such courses, enrollment remained consistently higher for males (MMM).

**Table 1.** Enrollment Statistics by Course, Timeframe, Gender, and Instructional Mode.

| Course | 2015 2016 2017 | | | 2018 2019 | | | 2020 2021 | | | Trend |
|---|---|---|---|---|---|---|---|---|---|---|
| | Female FtF | Male FtF | *n* | Female FtF | Male FtF | *n* | Female O | Male O | *n* | |
| Arabic Culture | 52.4% | 47.6% | 10,283 | 46.1% | 53.9% | 4980 | 45.8% | 54.2% | 6268 | FMM |
| Communication | 51.3% | 48.7% | 10,855 | 45.3% | 54.7% | 4610 | 51.3% | 48.7% | 4512 | FMF |
| Mathematics | 51.0% | 49.0% | 6224 | 48.1% | 51.9% | 4066 | 42.9% | 57.1% | 4061 | FMM |
| Natural Sciences | 44.5% | 55.5% | 4405 | 47.4% | 52.6% | 2966 | 45.5% | 54.5% | 2706 | MMM |
| Wellness | 53.8% | 46.2% | 5550 | 41.4% | 58.6% | 2755 | 53.4% | 46.6% | 3020 | FMF |
| Prof. Competencies | 53.8% | 46.2% | 7653 | 49.2% | 50.8% | 3594 | 48.7% | 51.3% | 5164 | FMM |
| Self-Assessment | 56.3% | 43.7% | 5734 | 47.6% | 52.4% | 3939 | 46.1% | 53.9% | 3675 | FMM |

Note: The term "trend" indicates significant shifts in enrollment between male and female students across time periods. The letter "F" or "M" is used to symbolize the greater enrollment numbers of either females or males.

Second, enrollment changes from face-to-face courses offered during the recent pre-pandemic period to online courses were examined (see Table 2), as they might indicate a different response of female and male students to education during the pandemic: fleeing or withstanding change. The recent pre-pandemic period was selected to include as much as possible students of the same academic cohorts. Thus, for each category of courses, a chi-squared test was performed to determine whether the enrollment rates for females and males varied as a function of the instructional mode (face-to-face and online). A Bonferroni correction was applied to avoid experiment-wise alpha, thereby treating as significant all tests with $p < 0.007$. Females had greater enrollment online than face-to-face, whereas males displayed the opposite pattern in Communication courses ($\chi^2(1, n = 9122) = 32.55, p < 0.001$) as well as Wellness courses ($\chi^2(1, n = 5775) = 82.64, p < 0.001$). In Math courses, males' enrollment was greater online than face-to-face, whereas females' enrollment was lower online than face-to-face ($\chi^2(1, n = 8127) = 21.82, p < 0.001$). To understand the available data, students' responses to end-of-course surveys were considered. Although gender differences were limited to a few categories of courses, they underscored gendered views of specific knowledge domains. Namely, a decline in Math online enrollment corresponded to females' frequently reported concerns regarding Math instruction, which the prospect of online delivery magnified. Increased online enrollment in Communication or Wellness courses matched females' sense of linguistic proficiency and preference for the Humanities over Math, as well as their keen interest in health matters.

In all other course categories, enrollment followed the same pattern for males and females ($\chi^2 < 2.16$, ns). Namely, enrollment tended to be higher online in Arabic Culture and Professional Competency courses, whereas it tended to be higher face-to-face in Self-Assessment and Natural Science courses. For the latter, the pattern reflected a common students' preference for in-person instruction due to their views of such courses as being challenging.

**Table 2.** Enrollment Statistics by Course, Gender, and Instructional Mode.

| Course | Female FtF | Female Online | *n* | Male FtF | Male Online | *n* |
|---|---|---|---|---|---|---|
| Arabic Culture | 44.4% | 55.6% | 5164 | 44.1% | 55.9% | 6084 |
| Communication * | 47.4% | 52.6% | 4401 | 53.4% | 46.6% | 4721 |
| Mathematics * | 52.9% | 47.1% | 3700 | 47.7% | 52.3% | 4427 |
| Natural Sciences | 53.3% | 46.7% | 2636 | 51.4% | 48.6% | 3036 |
| Wellness * | 41.4% | 58.6% | 2753 | 53.4% | 46.6% | 3022 |
| Prof. Competencies | 41.3% | 58.7% | 4283 | 40.8% | 59.2% | 4475 |
| Self-Assessment | 52.5% | 47.5% | 3569 | 51.1% | 48.9% | 4045 |
| | | | 26,506 | | | 29,810 |

Note: * significant shifts in enrollment between face-to-face (FtF) and online for male and female students.

*6.2. Assessment of Performance*

We also examined differences in performance (as measured by pass/fail grades) from face-to-face to online, as they might indicate whether indeed learning of female and male students was hurt during the pandemic. A passing grade corresponded to a letter grade of D+ (66%) or above. As the performance patterns of the face-to-face courses offered in the "remote pre-pandemic period" (six semesters including 2015, 2016, and 2017 academic years), replicated those of the face-to-face courses offered during the "recent pre-pandemic period" (three semesters immediately before the pandemic, including 2018 and 2019), the analyses combined the two periods. For each course, a chi-squared test was performed separately for females and males to determine whether pass/fail rates had changed from face-to-face to online instruction. A Bonferroni correction was applied to avoid experiment-wise alpha, thereby treating as significant all tests with $p < 0.004$.

Overall, female students were less likely to fail online in four of the seven GE courses selected for the present investigation (see Table 3). Among such courses, there was Math ($\chi^2(1, n = 6875) = 96.74, p < 0.001$), which yielded higher performance online notwithstanding concerns regarding the subject matter. The students provided several interpretations for this pattern of results in end-of-course surveys: (a) fears that Math courses would be more challenging online led them to enhance attention to and work harder on class activities, (b) extra time was available to work on Math problems since everyday mobility was curtailed and commuting eliminated, and (c) in-class experiences did not support expectations of unsurmountable difficulties. Similar explanations were given for Natural Science courses ($\chi^2(1, n = 4595) = 36.69, p < 0.001$), which were perceived as being challenging overall, irrespective of the mode of instruction. In contrast, Wellness and Arabic culture courses, which also illustrated better performance online ($\chi^2(1, n = 5737) = 70.71, p < 0.001$, and $\chi^2(1, n = 10,548) = 29.83, p < 0.001$, respectively), were reported as either more engaging or easier online.

The performance pattern of male students was checkered (see Table 2). As for female students, they were less likely to fail in online than face-to-face Math and Natural Science courses ($\chi^2(1, n = 7476) = 145.64, p < 0.001$, and $\chi^2(1, n = 5482) = 35.44, p < 0.001$, respectively), with similar reported reasons. However, in Arabic Culture and Wellness courses, males performed better face-to-face ($\chi^2(1, n = 10,983) = 41.70, p < 0.001$, and $\chi^2(1, n = 5588) = 30.28, p < 0.001$, respectively), whereas females performed better online. Males also performed better in face-to-face courses devoted to professional competencies ($\chi^2(1, n = 8014) = 188.43, p < 0.001$) and self-assessment ($\chi^2(1, n = 6553) = 6.83, p = 0.009$). For Arabic Culture, Wellness, and Professional Competency courses, online instruction was seen as less manageable and more dissatisfying. Self-assessment was perceived as harder online, mostly because of its requirement to build a personal portfolio for which feedback administered online was judged to be more cumbersome and less helpful. Male students might have had a more difficult time adapting to the online mode due to their having been the recipients

of prominence and privileges from the patriarchal society in which they, their parents, and grandparents were raised [36]. Although top-down gender equity forces have largely leveled the field for men and women, traces of past inequities may continue to exist [37–39], thereby making adaptation to changing circumstances difficult for those who have benefited from entitlement [43]. All the other categories of courses did not display differences between online and face-to-face for either females or males ($\chi s^2 \leq 3.96$, *ns*).

**Table 3.** Descriptive Statistics for Pass/Fail Grades by Course, Timeframe, and Instructional Mode for Male and Female Students.

| | 2015 2016 2017 | | 2018 2019 | | 2020 2021 | | |
|---|---|---|---|---|---|---|---|
| **Female Students** | **FtF** | | **FtF** | | **Online** | | **Higher Performance** |
| | Pass | Fail | Pass | Fail | Pass | Fail | |
| Arabic Culture * | 98.0% | 2.0% | 97.8% | 2.2% | 99.5% | 0.5% | Online |
| Communication | 97.7% | 2.3% | 97.6% | 2.4% | 97.9% | 2.1% | |
| Mathematics * | 91.4% | 8.6% | 91.2% | 8.8% | 98.2% | 1.8% | Online |
| Natural Sciences * | 94.0% | 6.0% | 93.9% | 6.1% | 98.3% | 1.7% | Online |
| Wellness * | 92.5% | 7.5% | 92.6% | 7.4% | 98.3% | 1.7% | Online |
| Prof. Competencies | 97.1% | 2.9% | 96.9% | 3.1% | 97.6% | 2.4% | |
| Self-Assessment | 93.7% | 6.3% | 93.8% | 6.2% | 95.0% | 5.0% | |
| **Male Students** | **FtF** | | **FtF** | | **Online** | | |
| | Pass | Fail | Pass | Fail | Pass | Fail | |
| Arabic Culture * | 97.1% | 2.9% | 98.1% | 1.9% | 95.0% | 5.0% | FtF |
| Communication | 92.2% | 7.8% | 93.2% | 6.8% | 91.8% | 8.2% | |
| Mathematics * | 90.9% | 9.1% | 90.2% | 9.8% | 98.3% | 1.7% | Online |
| Natural Sciences * | 96.1% | 3.9% | 96.5% | 3.5% | 99.3% | 0.7% | Online |
| Wellness * | 95.1% | 4.9% | 95.3% | 4.7% | 91.2% | 8.8% | FtF |
| Prof. Competencies * | 92.4% | 7.6% | 92.2% | 7.8% | 82.2% | 17.8% | FtF |
| Self-Assessment | 88.6% | 11.4% | 87.9% | 12.1% | 86.0% | 14.0% | FtF |

Note: * significant shifts in performance between face-to-face (FtF) and online.

*6.3. Forward Actions Grounded in Enrollment and Performance Data*

Our study is an exemplification of action research. It was initially motivated by instructors' informal concerns regarding learning during the pandemic, as well as by the extant literature suggesting that learners at the beginning of their educational endeavors (i.e., freshmen and sophomores) may be particularly susceptible to the changes in instruction brought about by the pandemic [5–7].

At the very minimum, the present study has encouraged instructors, eager to understand the unusual situation they were facing, to scrutinize the evidence provided by students in their classes, which may be a conduit to self-reflection and deeper questions regarding teaching and learning (i.e., localized approach). For instance, an instructor who had the habit of categorizing students' communications during office hours as either ordinary information-seeking behaviors regarding class activities or expressions of difficulties in adapting to the academic demands of college life (in addition to or without information-seeking behavior), was encouraged to compute an index of distress for semesters before and after the pandemic, by counting the number of interactions that reflected adaptation difficulties over the total number of interactions. Difficulties included being overwhelmed by the coursework, doubts about one's abilities, feelings of stress, a desire to quit, etc. The examination of the frequencies of the two classes of communications before and during the pandemic (with students merely identified by educational level) was intended to offer the instructor, colleagues, and academic counselors, opportunities for self-reflection and action to aid at-risk students. The evidence collected illustrated that although the records of adaptation difficulties tended to be more numerous for freshmen and sophomores than

juniors and seniors during the three semesters before the pandemic (11% versus 3%), they displayed very little change during the pandemic (14% versus 4%). Was the online mode limiting students' opportunities to express distress even during virtual office hours? If so, what features of the online mode contributed to students' reticence? Alternatively, were the students unexpectedly resilient in the face of sudden change? If so, what were the personal dispositions and environmental conditions responsible for resilience? The evidence collected brought about more questions than answers and highlighted the need to rely on additional sources of information to understand the emotional and social responses of students [32,33].

More broadly, the findings of our study regarding students' enrollment and academic performance not only offer faculty, administrators, and staff, opportunities for self-reflection and further investigation, but also dictate actions intended to benefit the population of students who contributed to the study [2]. Of course, enrollment and performance differences between the face-to-face and online modes are mere indices of students' responses to pandemic-related environmental changes that, at the very least, demand further examination (either localized or across the board) by faculty, administrators, and staff to determine their sources. At the selected institution, it is recognized that there are sources upon which the university has some degree of control, such as curriculum and instruction, and sources that predate academic admission (e.g., cognitive and affective dispositions, family's support of education, and gender stereotypes), for which control is either minimal or null. Students' end-of-course surveys and course reports, compiled by instructors, are routinely treated as valuable information on the sources of students' enrollment decisions and academic attainment, that can guide targeted interventions. Yet, data of action research such as ours can serve as a compass to direct the attention of stakeholders to particular areas in need of improvement. For instance, consider the lower academic success in the face-to-face classroom relative to the online classroom of GE courses involving Math and Natural Sciences. These courses are the foundations of careers in STEM fields (i.e., science, technology, engineering, and mathematics), which have only recently been opened to female students [44]. The relevance of STEM-related courses calls for an examination of what makes the academic demands of these courses offered synchronously online more attainable to students, even though objective examinations of course content and instruction by faculty point to the online and face-to-face modes as being largely equivalent. Namely, what are the psychological factors that facilitate students' attainment of learning outcomes in such courses when offered online? Are these factors pandemic-related or reflective of the online mode more broadly? Answers to these questions, which may rely on gathering information from focus groups in addition to course evaluations, can offer potentially useful information for restructuring face-to-face courses, since a return to normality is expected in the not too distant future. The obvious goal is to ensure a quality education for both female and male students.

At the selected institution, the quality of the education offered (including curricula presented through face-to-face and online instruction) is measured by six criteria: relevance, coherence, effectiveness, efficiency, predicted impact, and sustainability. Relevance concerns the significance of class activities to the students' daily lives. Coherence denotes the extent to which activities fit not only the course objectives but also students' needs. Effectiveness refers to whether the course objectives are achieved and students' needs are satisfied. Efficiency pertains to the extent to which course objectives and students' needs are satisfied in a trouble-free manner. Predicted impact refers to the extent to which class activities might have long-term and/or broader effects than those covered by the criterion of effectiveness. Sustainability refers to estimates regarding the permanence and usefulness of learning well beyond the confines of higher education. At the selected higher education institution, end-of-course surveys of female and male students, along with course reports compiled by instructors regarding class activities and students' outcomes in the face-to-face and online modes, are routinely reviewed by instructors and independent evaluators. Instruction being gender-segregated allows end-of-course surveys, which are

anonymous, to offer a clear-cut and simple window into female and male students' views and reactions to the courses in which they have enrolled. Each of the six criteria is used to measure the quality of a course on a 4-point scale including "developed", "developing", "under-developed", and "not available". The ratings are then used as a topic of discussion through focus groups and informal conversations, to determine the extent to which students' views of and reactions to the curriculum and the instruction imparted, merit attention and action. If a concern is identified, decisions are made on how instructors are to address it, either independently or in collaboration with other instructors, administrators, and/or counselors. Our data serve to enrich this practice by identifying areas of interest and/or concern.

Not surprisingly, preliminary evidence collected from students and instructors has led to the recognition of gender differences in responses to change, as well as interventions intended to ensure optimal learning. No clear-cut evidence has been uncovered of gender differences in learning strategies, including acquisition, codification, recuperation, and metacognition [45]. Rather, evidence has been found of females' greater ability to adapt to change, which is a property of emotional intelligence [46]. Adaptability appears to be fostered by an intense motivation to succeed academically to overcome past gender inequities and pursue independence. Interventions have encompassed changes in the way materials are posted and presented in class to enhance engagement, expansion of office hours to encourage participation, and greater in-class prodding of students' comments to promote empowerment in learning activities. Yet, the impact of each intervention on the academic success of male and female students continues to be measured through action research.

## 7. Conclusions

The pandemic may have challenged academic institutions around the world [47–50], but has also provided such institutions with opportunities to review established learning and teaching practices as well as to envision changes. The findings of the present investigation are consistent with those illustrating that in the presence of robust institutional support, students may not show any learning declines in the online mode (at least as measured by class grades) or may even display gains [20–23]. They add a focus on gender to the extant literature that examines the impact of the pandemic on academic success. Gender is a key demographic factor in a society in transition from a patriarchal system to one that fosters gender equity. The evidence-based approach adopted by the selected institution may lead to the identification of sources and the implementation of remedies that may not apply to other institutions. It may also benefit from a more fine-grained analysis of performance (e.g., letter grades rather than pass and fail outcomes). Most importantly, the collection of information regarding students' differences in attitudes and aptitudes may further inform the identification of remedies to teaching and learning difficulties that may be unique to the face-to-face and online modes, as well as the development of platforms that ameliorate online interactions [51]. For instance, studies have reported students' increased stress, anxiety, effort to self-regulate learning, feelings of isolation, and other difficulties during the pandemic [52], suggesting that the adaptation to online learning might have included not only challenges of a technological and instructional nature but also social and affective challenges [32,33]. Yet, studies have shown individual differences in resilience [32,53]. To this end, Collazos et al. [54,55] have noted that emotional awareness, which includes knowledge of people, task, and resources, as well as feedback intended to foster students' self-reflection and address undesirable emotional states [56], are important didactical tools for preserving a quality education during times of change.

Notwithstanding its content-related limitations, our study can serve as a simple model of evidence-based inquiry, employed by an academic institution whose goal is to provide quality education to all its students, face-to-face as well as online [57–59]. It brings to the forefront the theoretical, practical, and managerial implications of online instruction, and focuses the attention of stakeholders on the questions that remain objects of investigation

and debate in the extant literature. The theoretical implications encompass the optimal instructional conditions for ensuring deep learning [60] and students' dispositions that make it possible, irrespective of the online or face-to-face mode of instruction [32]. The practical implications entail a data-driven approach to teaching and learning adopted by instructors, staff, and administrators to guide their actions. The common goal is to ensure deep learning of valuable content, which can adequately prepare students for the careers of their choice, as well as foster an appreciation for learning as a life-long enterprise that does not stop at the time degrees are attained [61]. The managerial implications involve administrators, who not only coordinate the actions of instructors and staff to ensure the implementation of agreed-upon actions and verify the attainment of goals, but also can envision the means to foster a sustainable education. To this end, leadership needs to suit the cultural norms of the society in which an educational institution exists. For instance, in the Arab world, effective leadership in educational endeavors is considered one that relies on consultation (shura), role modeling, and a holistic approach that integrates the physical, cognitive, emotional, and motivational needs of students [62]. In a globalized world, differences among people are as important as similarities. They determine how information is selected and interpreted for consumption and acted upon in daily lives. A sensible understanding of how differences and similarities can contribute to the quality of the education offered by a given institution is key to its sustainability [59].

**Author Contributions:** All authors contributed equally to the research. Conceptualization, H.M.A., M.A.E.P. and O.J.E.-M.; methodology, H.M.A., M.A.E.P. and O.J.E.-M.; formal analysis, H.M.A., M.A.E.P. and O.J.E.-M.; data curation, H.M.A., M.A.E.P. and O.J.E.-M.; writing—original draft preparation, H.M.A., M.A.E.P. and O.J.E.-M.; writing—review and editing, H.M.A., M.A.E.P. and O.J.E.-M.; project administration, H.M.A., M.A.E.P. and O.J.E.-M. All authors have read and agreed to the published version of the manuscript.

**Funding:** This research received no external funding.

**Institutional Review Board Statement:** Research conducted under the purview of the Deanship of Research (exempt).

**Informed Consent Statement:** Not applicable.

**Data Availability Statement:** Data available upon request.

**Acknowledgments:** We are grateful to Emaan Nazeeruddin for her invaluable assistance.

**Conflicts of Interest:** The authors declare no conflict of interest.

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
