# Peer review of "Measuring the Impact of the Pandemic on Female and Male Students’ Learning in a Society in Transition: A Must for Sustainable Education"

_sustainability, doi:10.3390/su14063148_

Round 1

Reviewer 1 Report

Paper is well written, describing in a good manenr a problematic situation. However, it could be adequate to include some aspects:

  • Include related work section, in order to analyze similar works and main differences. In that way there are some works could be adeqauet to analyze and include: Collazos et al, The Use of e-Learning Platforms in a Lockdown Scenario—A Study in Latin American Countries
  • Analyze aspects like emotions and awareness (Collazos et al, Designing Online Platforms Supporting Emotions and Awareness)
  • Describe the recruitment process and how the study (methodology) could be replied
  • Improve discussion of main results
  • Improve conclusions and further work

Reviewer 2 Report

The subject of this paper is important and very interesting. Congratulation for taking up this subject. The studied sample and the set of data is really impressive. 

However, the paper suffers from quite severe methodological errors.  The Authors compare data  from a year (by the way: which year, exactly) before pandemic (with normal face-to-face education) and during pandemic (again: which year) and notice  essential statistical  evidence that some changes occured.  But, it seems that differences between ANY years could be significantly different. This should be a basis for further research: first to present data from several preceding year, then establish their typical/avarage variance, and only then to include the pandemic-year data and to analyze the impact of the pandemic. In my opinion, without adding these data the results of this paper are meaningless.

Other issues to be addressed:

lines 57-59: a probably but unverified hypothesis is put forward (that freshmen are more susceptible to changes in instructions). Why do not present some data supporting it? 

Table 1: please provide data for several years preceding the pandemic

Table 2: are the presented fail/pass ratios stable in preceding years? The data must be provided.  

Which years are considered?  Where online students aware of this online mode before their enrollment or after it? 

In general, the paper contains too much of rather meaningless general discussions (a large part of the Conclusion goes in the same direction) and too few concrete results and analyses.

Reviewer 3 Report

Thank you very much for the opportunity to read this interesting paper. There are some lacks that can be improved to add value to the research.

Maybe you can connect the research to the social learning theory to have a ground theory.

Missing the source: This mindset has led to the argument that “well-planned online learning experiences are meaningfully different from courses offered online in response to a crisis or disaster” (p. 1).

I miss some objectives or some hypothesis, maybe some objectives or hypotheses can and value to the research to a better flow of the research.

Also, maybe some additional test can add value because now it is a descriptive statistic.

The conclusion must be restructured and extended of theoretical, practical, and managerial implication. And a part of limitations can be added.

I wish you good luck!

Round 2

Reviewer 2 Report

In my first report I wrote in my conclusion:

"In general, the paper contains too much of rather meaningless general discussions (a large part of the Conclusion goes in the same direction) and too few concrete results and analyses". 

After revision it is even worse. No new concrete results or analyses (of the results), many new words which do not add new insights.

The Authors explained why fresmen and sophomores are "more susceptible for changes" in general terms, but I still would prefer to see the difference in the actual data. However, this is a minor point.

The crucial issue is the lack of any comparable data for period before and during pandemic.  In the revised version the Authors mention that their studies include 3 semesters of face-to-face education and 3 semesters of online courses.  However, probably they summed/averaged results of all these semesters into two groups:  face-to-face and online.  The Authors write in their response: "Thus, we are unable to provide data for online courses before the pandemic". But this is not demanded by me. I just would like to see the data for several years of standard face-to-face education, 3 semesters (to some extent) could be sufficient.

My point, presented in my first report, was to see an empirical variance of the results (marks) in a normal situation, how they changes from year to year, from semester to semester. 

Without that the presented results are meaningless.

I recommend to reject the paper, enecouraging resubmission after the data (for several semesters at least) will be suitably presented and interpreted, and a new, better paper will be submitted. 

Reviewer 3 Report

Thank you for the revised version. There are some few issues like arranging the text.

Good Luck!
